# On the calculation of normalized viscous-plastic sea ice stresses

Jean-François Lemieux [1] and Frédéric Dupont [2]

[1]Recherche en Prévision Numérique Environnementale/Environnement et Changement Climatique Canada, 2121 route Transcanadienne, Dorval, Qc, Canada.
[2]Service Météorologique Canadien, Environnement et Changement Climatique Canada, 2121 route Transcanadienne, Dorval, Qc, Canada.

**Correspondence:** J-F. Lemieux (jean-francois.lemieux@canada.ca)

**Abstract.** Calculating and plotting the normalized states of stress for viscous-plastic sea ice models is a common diagnostic for evaluating the numerical convergence and the physical consistency of a numerical solution. Researchers, however, usually do not explain how they calculate the normalized stresses. Here, we argue that care must be taken when calculating and plotting the normalized states of stress. A physically consistent and numerically converged solution should exhibit normalized stresses that are inside (viscous) or on (plastic) the normalized yield curve. To do so, two possible mistakes need to be avoided. First, when using an implicit solver, normalized stresses should be computed from viscous coefficients and replacement pressure calculated using the previous numerical iterate and the strain rates at the numerator calculated from the latest iterate. Calculating the stresses only from the latest iterate falsely indicates that the solution has numerically converged. Second, for both implicit and explicit (i.e., the EVP) solvers, the stresses should be normalized by the ice strength and not by the replacement pressure. Using the latter, normalized states of stress only lie on the yield curve (i.e., falsely indicating there are no viscous states of stress).

## 1 Introduction

Sea ice deformations, associated with the formation of leads, pressure ridges and shear lines, strongly influence the evolution of the sea ice cover in both polar oceans. As they affect the thickness distribution, sea ice deformations have an important impact on the exchange of heat, moisture and momentum between the atmosphere and the underlying ocean. To properly represent these processes in a model, it is essential that rheology, i.e. the relation between applied stresses, material properties and resulting deformations is correctly formulated.

Although some authors have recently proposed new sea ice rheologies (e.g., Girard et al. (2011)), most sea ice models are still based on the viscous-plastic (VP) formulation introduced by Hibler (1979). With the VP rheology, the ice is treated as a very viscous fluid (creep flow) when the internal stresses are small. However, once the stresses reach critical values defined by a yield curve, the ice flows as a plastic material and large deformations (i.e., large spatial gradients of the velocity field) can occur.

Calculating and plotting the normalized states of stress with respect to the normalized yield curve is a useful diagnostic for assessing the physical consistency and numerical convergence of a VP solution. Indeed, this method can confirm whether a

25 sea ice rheology is properly implemented in a model. The method is also helpful for evaluating numerical convergence. This is especially true for the explicit elastic-VP (EVP) solver (e.g., Hunke (2001)) which does not include a measure of convergence such as a residual. Unfortunately, researchers usually do not explain how they calculate this diagnostic (e.g. Zhang and Hibler (1997); Hunke (2001); Lemieux and Tremblay (2009); Wang and Wang (2009); Kimmritz et al. (2015)). As demonstrated here, care must be taken when calculating the normalized stresses as two potential mistakes could lead to a misinterpretation

of modeling results. The purpose of this manuscript is to provide a short guide on how to calculate and plot the normalized states of stress for assessing physical consistency and convergence of numerical solutions.

## 2 The viscous-plastic sea ice rheology

With the Hibler (1979) VP rheology, the components $\sigma_{ij}$ of the stress tensor are given by

$$\sigma_{ij} = 2\eta\dot{\epsilon}_{ij} + [\zeta - \eta]\dot{\epsilon}_{kk}\delta_{ij} - P_p\delta_{ij}/2, \quad i,j = 1,2, \tag{1}$$

where $\delta_{ij}$ is the Kronecker delta, $\dot{\epsilon}_{ij}$ are the strain rates defined by $\dot{\epsilon}_{11} = \frac{\partial u}{\partial x}$, $\dot{\epsilon}_{22} = \frac{\partial v}{\partial y}$ and $\dot{\epsilon}_{12} = \frac{1}{2}(\frac{\partial u}{\partial y} + \frac{\partial v}{\partial x})$ with $u$ and $v$ the components of the horizontal sea ice velocity vector, $\dot{\epsilon}_{kk} = \dot{\epsilon}_{11} + \dot{\epsilon}_{22}$, $\zeta$ is the bulk viscosity, $\eta$ is the shear viscosity and $P_p$ is the ice strength (we follow the notation of Kreyscher et al. (2000)).

The formulation of the viscosities depends on the yield curve and the flow rule. In the following, $\zeta$ and $\eta$ are based on the widely used elliptical yield curve with a normal flow rule (Hibler, 1979):

$$\zeta = \frac{P_p}{2\Delta}, \tag{2}$$

$$\eta = \zeta e^{-2}, \tag{3}$$

where $\Delta = \left[(\dot{\epsilon}_{11} + \dot{\epsilon}_{22})^2 + e^{-2}(\dot{\epsilon}_{11} - \dot{\epsilon}_{22})^2 + 4e^{-2}\dot{\epsilon}_{12}^2\right]^{\frac{1}{2}}$, and $e$ is the aspect ratio of the ellipse, i.e. the ratio of the long and
45 short axes of the elliptical yield curve.

When $\Delta$ tends toward zero, equations (2) and (3) become singular. To avoid this problem, Hibler (1979) proposed to limit the maximum values of viscosities which is equivalent to limiting the minimum value of $\Delta$. Hence, $\zeta$ is expressed as

$$\zeta = \frac{P_p}{2\Delta^*}, \tag{4}$$

where $\Delta^* = \max(\Delta, \Delta_{min})$ with $\Delta_{min} = 2 \times 10^{-9}$ s$^{-1}$. Note that other approaches for limiting the viscous coefficients have been proposed (e.g., Kreyscher et al. (2000); Lemieux and Tremblay (2009)).

A drawback of the standard VP rheology is that the term $-P_p \delta_{ij}/2$ in equation (1) can cause the ice to deform even in the absence of forcing. To remedy this problem, $-P_p \delta_{ij}/2$ is replaced by $-P\delta_{ij}/2$, where $P$ is a function of the strain rates. The simplest formulation of $P$ is

$$P = P_p \frac{\Delta}{\Delta^*},$$ (5)

where $P$ tends toward zero for small deformations while it tends toward $P_p$ for large deformations.

$P$ is sometimes referred to as the replacement pressure (e.g., Hunke and Lipscomb (2010)). The use of a replacement method such as the one described above is now widely used in VP sea ice models (e.g., Wang and Wang (2009); Losch et al. (2010); Hunke and Lipscomb (2010)).

## 3 The normalized yield curve

Using equations (1), (3), (4), (5) and the definition of $\Delta$, one can obtain

$$P_p^2 \left(\frac{\Delta}{\Delta^*}\right)^2 = [\sigma_{11} + \sigma_{22} + P]^2 + e^2 \left[(\sigma_{11} - \sigma_{22})^2 + 4\sigma_{12}^2\right].$$ (6)

Introducing the principal stresses $\sigma_{p1}$ and $\sigma_{p2}$ given by

$$\sigma_{p1}, \sigma_{p2} = \frac{\sigma_{11} + \sigma_{22}}{2} \pm \sqrt{\left(\frac{\sigma_{11} - \sigma_{22}}{2}\right)^2 + \sigma_{12}^2},$$ (7)

equation (6) becomes

$$P_p^2 \left(\frac{\Delta}{\Delta^*}\right)^2 = [\sigma_{p1} + \sigma_{p2} + P]^2 + e^2 \left[(\sigma_{p1} - \sigma_{p2})^2\right].$$ (8)

As demonstrated below, the correct way to normalize the stresses in equation (8) is to divide them by the ice strength $P_p$ which leads to

$$\left(\frac{\Delta}{\Delta^*}\right)^2 = \left[\frac{\sigma_{p1} + \sigma_{p2} + P}{P_p}\right]^2 + e^2 \left[\frac{\sigma_{p1} - \sigma_{p2}}{P_p}\right]^2.$$ (9)

Defining $\sigma_{p1}^n = \sigma_{p1}/P_p$ and $\sigma_{p2}^n = \sigma_{p2}/P_p$, we obtain

$$\left(\frac{\Delta}{\Delta^*}\right)^2 = \left[\sigma_{p1}^n + \sigma_{p2}^n + \frac{P}{P_p}\right]^2 + e^2\left[\sigma_{p1}^n - \sigma_{p2}^n\right]^2, \tag{10}$$

which describes a family of ellipses that depend on the ratio $\Delta/\Delta^*$ for their size and on the ratio $P/P_p$ for their center. Equation (10) with $\Delta/\Delta^* = P/P_p = 1$ defines what we refer to as the normalized yield curve in principal stress space. Hence, according to our rheology, normalized plastic stresses should fall on the normalized yield curve while normalized viscous stresses should lie on smaller ellipses inside the normalized yield curve (Geiger et al., 1998).

## 4 Experimental setup

The divergence of the stress tensor (described in section 2), that is $\nabla \cdot \sigma$, is one of the terms of the sea ice momentum equation. The momentum equation is discretized in space and in time (see for example Lemieux et al. (2012) for details). It is either solved implicitly with a Picard solver (e.g. Zhang and Hibler (1997); Losch et al. (2010)) or with a Newton solver (e.g. Lemieux et al. (2012); Losch et al. (2014); Mehlmann and Richter (2017)) or it is solved explicitly with the EVP approach (Hunke, 2001) or using the modified EVP with pseudo-time stepping (e.g. Kimmritz et al. (2015)).

The numerical simulations for this paper were conducted with the Picard solver of the McGill sea ice model (see Lemieux and Tremblay (2009) for details). The spatial resolution is 10 km and the time step is 30 min. All the experiments with the elliptical yield curve were done with the ice strength parameter $P^*$ set to $27.5 \times 10^3$ Nm$^{-2}$ and $e$=2. The model was restarted on January $1^{st}$ 2002 12 UTC from a long-term simulation. The states of stress were calculated from solutions obtained at the first time level (i.e., 12h30 UTC), We will discuss later how our conclusions apply to the other types of solvers.

With a Picard solver, one has to solve a nonlinear system of equations that can be concisely written as $\mathbf{A}(\mathbf{u})\mathbf{u} = \mathbf{b}(\mathbf{u})$ where $\mathbf{u}$ is a vector that contains all the $u$ and $v$ velocity components on the grid, $\mathbf{A}$ is a sparse matrix and $\mathbf{b}$ is a vector that contains terms such as the atmospheric stress. It is important to mention that the elements of the matrix $\mathbf{A}$ depend on the viscous coefficients $\zeta$ and $\eta$ and that the vector $\mathbf{b}$ contains the replacement pressure $P$. Implicit solvers such as Picard solve a series of linearized systems of equations in order to find the solution $\mathbf{u}$ of the nonlinear system of equations. This algorithm can be expressed as

```
1. Start with an initial iterate u⁰
do k = 1, k_max
    2. Solve A(u^(k-1))u^k = b(u^(k-1)) with a linear solver
    3. Stop if ||F(u^k)|| < γ_nl ||F(u⁰)||
```

```
enddo
```

where $\mathbf{F}(\mathbf{u}^k) = \mathbf{A}(\mathbf{u}^k)\mathbf{u}^k - \mathbf{b}(\mathbf{u}^k)$ is the residual at iteration $k$, the symbol $|| \; ||$ denotes the l2-norm and $\gamma_{nl} < 1$ is the non-linear convergence parameter. The iterations of this loop are referred to as nonlinear iterations or as in Lemieux and Tremblay (2009) as outer loop iterations. A 'fully' converged solution for $\mathbf{u}$ is characterized by a very small residual ($\gamma_{nl}$ needs to be set to a value $\ll 1$). As the stresses are function of $\mathbf{u}$, a 'fully' converged velocity vector leads to states of stress that are either on

(plastic) or inside (viscous) the yield curve.

In order to shorten the manuscript, the presentation of the algorithm above has been simplified. For numerical stability, the water stress should be linearized with $(\mathbf{u}^{k-1} + \mathbf{u}^{k-2})/2$ (Hibler and Ackley, 1983). Lemieux and Tremblay (2009) also linearized the rheology term with $(\mathbf{u}^{k-1} + \mathbf{u}^{k-2})/2$. For faster convergence of the Picard solver, we recommend to use

$(\mathbf{u}^{k-1} + \mathbf{u}^{k-2})/2$ only for the water stress and to linearize the rheology term with $\mathbf{u}^{k-1}$. Hence, it is important to notice that when linearizing the system of equation (in step 2), $\zeta$, $\eta$ and $P$ are expressed as a function of $\mathbf{u}^{k-1}$.

## 5 The calculation of normalized states of stress

The steps for calculating and plotting the normalized stresses are given below.

```
1. Solve the nonlinear system of equations for uᵏ ∼ u
```
2. Calculate $\sigma_{ij} = 2\eta(\mathbf{u}^{k-1})\dot{\epsilon}_{ij}(\mathbf{u}^k) + [\zeta(\mathbf{u}^{k-1}) - \eta(\mathbf{u}^{k-1})]\dot{\epsilon}_{kk}(\mathbf{u}^k)\delta_{ij} - P(\mathbf{u}^{k-1})\delta_{ij}/2,$     i,j=1,2
3. Calculate $\sigma_{p1}^n, \sigma_{p2}^n = \frac{\sigma_{11}+\sigma_{22}}{2P_p} \pm \frac{1}{P_p}\sqrt{\left(\frac{\sigma_{11}-\sigma_{22}}{2}\right)^2 + \sigma_{12}^2}$
```
4. Plot the σⁿₚ₁,σⁿₚ₂ using symbols such as circles
```

5. Plot the normalized yield curve $\left[\sigma_{p1}^n + \sigma_{p2}^n + 1\right]^2 + e^2\left[\sigma_{p1}^n - \sigma_{p2}^n\right]^2 = 1$ as a reference

where the calculations in steps 2 and 3 should be done for all the ice covered grid cells (here grid cells with a concentration larger than 0.5 are considered). The $\sigma_{ij}$ (step 2) and $\sigma_{p1}^n, \sigma_{p2}^n$ (step 3) are calculated so that they are collocated at the tracer point of our model C-grid. Step 2 should be omitted for the standard and modified EVP; the time-stepped stresses should be

used directly for step 3. Note that normalized stresses can also be plotted using the stress invariants $\sigma_I = (\sigma_{p1} + \sigma_{p2})/2$ and $\sigma_{II} = (\sigma_{p1} - \sigma_{p2})/2$.

Following this method allows one to assess the physical consistency and the numerical convergence of the solution. We mean by physical consistency and numerical convergence that the states of stress are at their final position inside (viscous) or

135 on (plastic) the yield curve. Many authors (e.g., Zhang and Hibler (1997); Lemieux and Tremblay (2009)) have indeed shown that an approximate solution that has not sufficiently converged exhibits unrealistic states of stress that are outside the yield

curve. This is shown in Fig. 1. For two (Fig. 1a) or 10 nonlinear iterations (Fig. 1b), the approximate solution has not converged and shows unrealistic states of stress. The fully converged solution (Fig. 1c) demonstrates physical consistency and numerical convergence. The fully converged solution was obtained by setting $\gamma_{nl}$ to $1 \times 10^{-8}$. Note that, in general, the fact that states of stress are on or inside the yield curve does not imply full convergence; the final positions (on and inside the yield curve) are obtained once $\mathbf{u}^k$ is the fully converged solution (Lemieux and Tremblay, 2009).

Two mistakes need to be avoided in order to obtain similar results as in Fig. 1 and therefore to be able to evaluate the numerical convergence of the solution and physical consistency.

First, one has to consider the way the nonlinear system of equations is solved. It is crucial to note that the $\sigma_{ij}$ in step 2 should be calculated from $\zeta$, $\eta$ and $P$ that are a function of the previous iterate $\mathbf{u}^{k-1}$ and the strain rates at the numerator from the latest iterate $\mathbf{u}^k$. Let's consider that one calculates the stresses only based on the latest iterate $\mathbf{u}^k$, that is the viscous coefficients $\zeta$ and $\eta$ and the replacement pressure are functions of $\mathbf{u}^k$ instead of $\mathbf{u}^{k-1}$. Fig. 2 shows the normalized states of stress that are obtained in this case after only two nonlinear iterations. One might conclude from this figure that the solution has converged as all the states of stress appear to be VP while we know this is not the case from Fig. 1a. This is important because a "true" converged solution exhibits better defined sea ice leads (and deformations, Lemieux and Tremblay (2009)), where large moisture/energy/salt fluxes are present between the sea ice, the ocean and the atmosphere.

This apparent numerical convergence of the solution is a consequence of the use of a rate-independent plastic rheology. This can be easily understood by considering a 1D VP example. Assuming that sea ice does not have tensile strength and that it exhibits a large convergent deformation, the 1D relation between the stress ($\sigma$) and the deformation ($\dot{\epsilon} = \frac{\partial u}{\partial x}$) is given by

$$\sigma = \zeta \dot{\epsilon} - \frac{P}{2}, \tag{11}$$

where $\zeta = \frac{P_p}{2|\dot{\epsilon}|}$ and $P = P_p$ for a large plastic deformation.

Correctly expressing $\zeta$ as a function of $\mathbf{u}^{k-1}$ and $\dot{\epsilon}$ as a function of $\mathbf{u}^k$ (with $\dot{\epsilon}^k = \dot{\epsilon}(\mathbf{u}^k)$), we obtain

$$\sigma = \frac{P_p}{2|\dot{\epsilon}^{k-1}|} \dot{\epsilon}^k - \frac{P_p}{2}, \tag{12}$$

which is equal to $-P_p$ only once the numerical solution has converged.

On the other hand, expressing both $\zeta$ and $\dot{\epsilon}$ as a function of $\mathbf{u}^k$ leads to

$$\sigma = \frac{P_p}{2|\dot{\epsilon}^k|} \dot{\epsilon}^k - \frac{P_p}{2}, \tag{13}$$

which is always equal to $-P_p$ whatever the velocity field $\mathbf{u}^k$ used.

A second possible mistake would be to normalize the principal stresses in step 3 with the replacement pressure $P$ instead of using $P_p$. Indeed, dividing equation (8) by $P^2$, we get

$$
1 = \left[ \frac{\sigma_{p1} + \sigma_{p2} + P}{P} \right]^2 + e^2 \left[ \frac{\sigma_{p1} - \sigma_{p2}}{P} \right]^2. \tag{14}
$$

Defining $\sigma_{p1}^n = \sigma_{p1}/P$ and $\sigma_{p2}^n = \sigma_{p2}/P$, we obtain

$$
1 = \left[ \sigma_{p1}^n + \sigma_{p2}^n + 1 \right]^2 + e^2 \left[ \sigma_{p1}^n - \sigma_{p2}^n \right]^2, \tag{15}
$$

which is the equation of an ellipse with a size and a center that are fixed. Equation (15) is in fact the same equation as the one for the normalized yield curve (i.e., equation (10) with $\Delta/\Delta^* = P/P_p = 1$). Simulated stresses normalized by $P$ indeed converge toward this fixed ellipse. This is shown in Fig. 3 for two (a), 10 (b) and the fully converged solution (c). The converged normalized states of stress do not exhibit a realistic solution as all the stresses appear to be plastic.

## 6  Broader considerations

The recommendations given above remain the same if another approach is used for limiting the viscous coefficients (see equation 4). Numerical experiments with the approach of Kreyscher et al. (2000) or with the hyperbolic tangent of Lemieux and Tremblay (2009) allow one to draw the same conclusions (not shown).

While it is not recommended to linearize the rheology term with the previous two iterates (as done by Lemieux and Tremblay (2009)), the stresses in step 2 (see beginning of section 5) should in this case be obtained from $\sigma_{ij} = 2\eta(\mathbf{u}_l)\dot{\epsilon}_{ij}(\mathbf{u}^k) + [\zeta(\mathbf{u}_l) - \eta(\mathbf{u}_l)]\dot{\epsilon}_{kk}(\mathbf{u}^k)\delta_{ij} - P(\mathbf{u}_l)\delta_{ij}/2$ with $\mathbf{u}_l = (\mathbf{u}^{k-1} + \mathbf{u}^{k-2})/2$.

If one does not use a replacement pressure, the stresses in step 2 should be calculated the same way with $P = P_p$. Instead of lying on ellipses defined by equation (10), the normalized viscous states of stress would lie on concentric ellipses centered at $\sigma_{p1}^n = \sigma_{p2}^n = -0.5$ (Geiger et al., 1998).

As demonstrated below, our recommendations also apply when using other yield curves in a VP framework. As an example, additional numerical experiments were conducted with a Mohr-Coulomb yield curve with compressive capping (Ip et al., 1991). This different constitutive law is obtained by expressing the viscous coefficients and the replacement pressure as

$$\zeta = \frac{P_p}{2|\dot{\epsilon}_I^*|}, \tag{16}$$

$$P = 2\zeta|\dot{\epsilon}_I|, \tag{17}$$

$$\eta = \frac{\left(\frac{P}{2} - \zeta\dot{\epsilon}_I\right)\sin\phi}{2\dot{\epsilon}_s^*} \tag{18}$$

where $\dot{\epsilon}_I = \dot{\epsilon}_{11} + \dot{\epsilon}_{22}$ is the divergence, $|\dot{\epsilon}_I^*| = \max(|\dot{\epsilon}_I|, d_{min})$ with $d_{min}$ a small deformation similar to $\Delta_{min}$, $\phi$ is the angle of friction, $\dot{\epsilon}_s^* = \max(\dot{\epsilon}_s, s_{min})$ with $\dot{\epsilon}_s = \left[\left(\frac{\dot{\epsilon}_{11} - \dot{\epsilon}_{22}}{2}\right)^2 + \dot{\epsilon}_{12}^2\right]^{1/2}$ the maximum shear strain rate and $s_{min}$ another small deformation, here set equal to $d_{min}$. In terms of the stress invariants, the Mohr-Coulomb failure criterion is simply written as $\sigma_{II} = -\sigma_I \sin\phi$. This Mohr-Coulomb implementation assumes a pure shear flow rule. Divergence (larger than $d_{min}$) can only occur at the tip of the triangle and convergence when $\sigma_I = -P_p$.

It is observed that with this new rheology, the Picard solver really struggles to obtain a numerically converged solution. With $P^* = 27.5 \times 10^3$ Nm$^{-2}$, $d_{min} = 2 \times 10^{-9}$ s$^{-1}$ and $sin\phi = 0.5$ (i.e., $\phi = 30°$), the solver does not converge. When calculating the normalized stresses the correct way (as in step 2 in section 5), there are states of stress outside the yield curve (not shown). However, similar to the results obtained with the elliptical yield curve (see Fig.2), the normalized stresses (shown in Fig.4 in stress invariant space) after two nonlinear iterations appear to have converged if only $\mathbf{u}^k$ is used for calculating the $\sigma_{ij}$.

To obtain a fully converged solution (with $\gamma_{nl} = 1 \times 10^{-8}$), $P^*$, $d_{min}$ and $sin\phi$ were respectively set to $5 \times 10^2$ Nm$^{-2}$, $1 \times 10^{-8}$ s$^{-1}$ and 0.01. Consistent with the results obtained with the ellipse, the converged stresses normalized by $P_p$ are either on or inside the yield curve (not shown). Again, normalizing the converged stresses by the replacement pressure falsely indicates there are no stresses in the viscous regime (Fig. 5). Strangely, there are no states of stress on the long side of the triangle; all the states of stress appear to be at the tip and the short side of the triangle. This can be easily understood by using equations (16) and (17) to calculate the normalized first stress invariant

$$\sigma_I^n = \frac{\sigma_I}{P} = \frac{\zeta\dot{\epsilon}_I}{P} - \frac{P}{2P} = \frac{\dot{\epsilon}_I}{2|\dot{\epsilon}_I|} - \frac{1}{2}, \tag{19}$$

which is equal to zero (tip of the triangle) when $\dot{\epsilon}_I > 0$ and equal to -1 (short side of the triangle) when $\dot{\epsilon}_I < 0$.

## 7 Conclusion

We have described how the normalized states of stress should be calculated and plotted in order to assess the numerical convergence and physical consistency of a VP solution. To do so, modelers should avoid two possible mistakes.

First, to evaluate the numerical convergence of an approximate solution, one should calculate stresses from viscous coefficients and replacement pressure that are a function of the previous iterate $\mathbf{u}^{k-1}$ and the strain rates at the numerator from the latest iterate $\mathbf{u}^{k}$. This conclusion applies to all implicit solvers. As the EVP and modified EVP approaches include time-stepping equations for the stresses, one simply needs to calculate the normalized stresses from the stress outputs. This issue of misinterpretation of numerical convergence with normalized stresses is therefore more prone to occur with Picard and Newton solvers.

Second, the stresses should be normalized by the ice strength; not by the replacement pressure. Using the latter causes all the normalized stresses to lay on the normalized yield curve, falsely indicating there are no stresses in the viscous regime. This issue can affect the implicit solvers but also the EVP and modified EVP approaches.

This manuscript should serve as a guide on how to calculate and plot normalized VP states of stress for assessing physical consistency and convergence of numerical solutions. It also complements and gives more details about one of the sea ice diagnostics suggested for the CMIP6 sea-ice intercomparison project (Notz et al., 2016).

*Code availability.* Revision 333 of the McGill sea ice model, with some modifications for including the Mohr-Coulomb rheology, was used for the numerical experiments described in this manuscript. The code is available on Zenodo at http://doi.org/10.5281/zenodo.3629542. The Zenodo deposit also includes the output files for the normalized stresses and the Matlab routine used for plotting.

*Author contributions.* JFL and FD derived the equations and designed the numerical experiments. JFL ran the experiments and wrote the manuscript.

*Competing interests.* JFL and FD do not have any competing interests.

*Acknowledgements.* We thank Philippe Blain, Amélie Bouchat, Bruno Tremblay and two anonymous reviewers for their helpful comments about this manuscript.

245

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

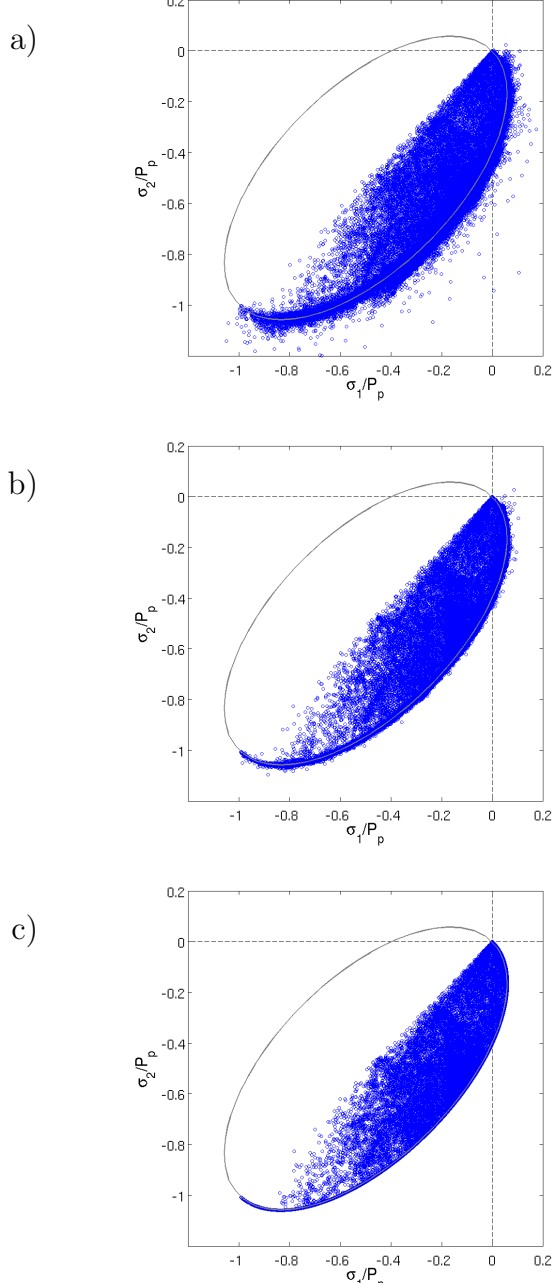

**Figure 1.** Principal stresses normalized by the ice strength $P_p$ after two (a), 10 (b) nonlinear iterations and the fully converged solution (c).

$$\left[\sigma_{p1}^n + \sigma_{p2}^n + 1\right]^2 + e^2 \left[\sigma_{p1}^n - \sigma_{p2}^n\right]^2 = 1$$ with $e = 2$ is the normalized yield curve (solid black line).

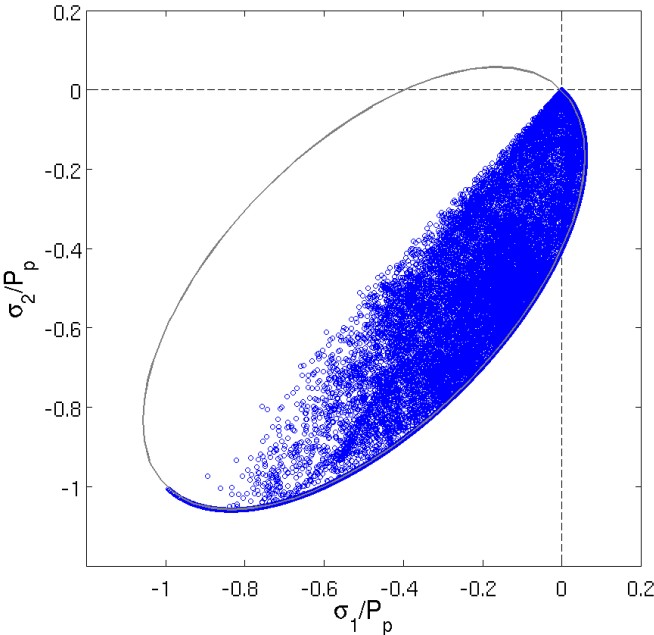

**Figure 2.** Principal stresses after two nonlinear iterations calculated only from $\mathbf{u}^k$ and normalized by the ice strength $P_p$. The solution appears to be numerically converged because the $\sigma_{ij}$ are only a function of $\mathbf{u}^k$. $\left[\sigma_{p1}^n + \sigma_{p2}^n + 1\right]^2 + e^2\left[\sigma_{p1}^n - \sigma_{p2}^n\right]^2 = 1$ with $e = 2$ is the normalized yield curve (solid black line).

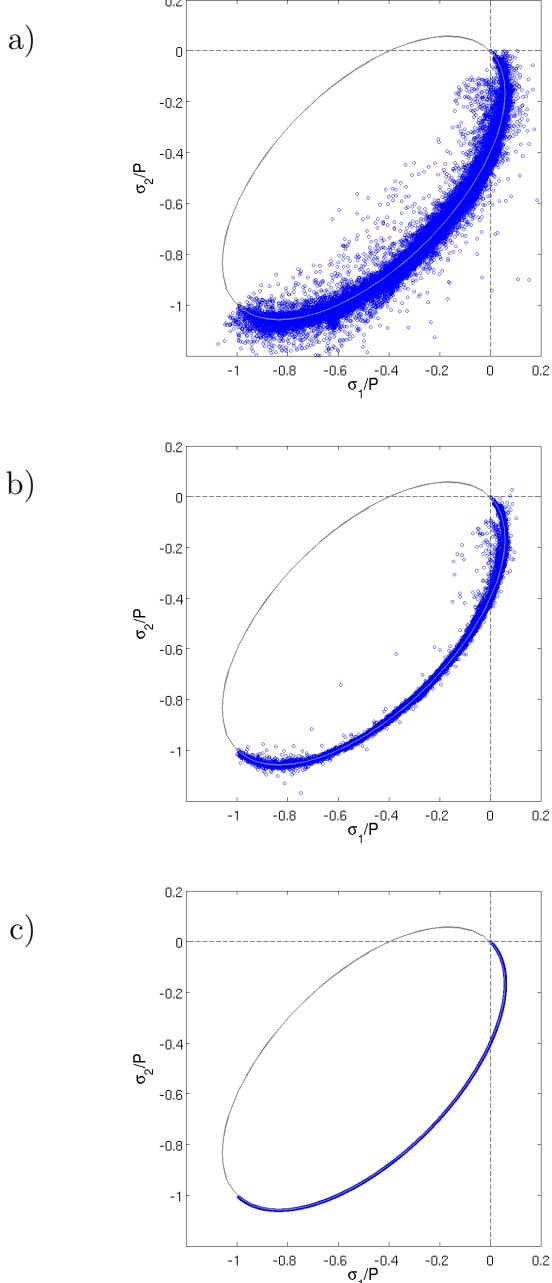

**Figure 3.** Principal stresses normalized by the replacement pressure $P$ after two (a), 10 (b) nonlinear iterations and the fully converged solution (c). $\left[\sigma_{p1}^n + \sigma_{p2}^n + 1\right]^2 + e^2 \left[\sigma_{p1}^n - \sigma_{p2}^n\right]^2 = 1$ with $e = 2$ is the normalized yield curve (solid black line).

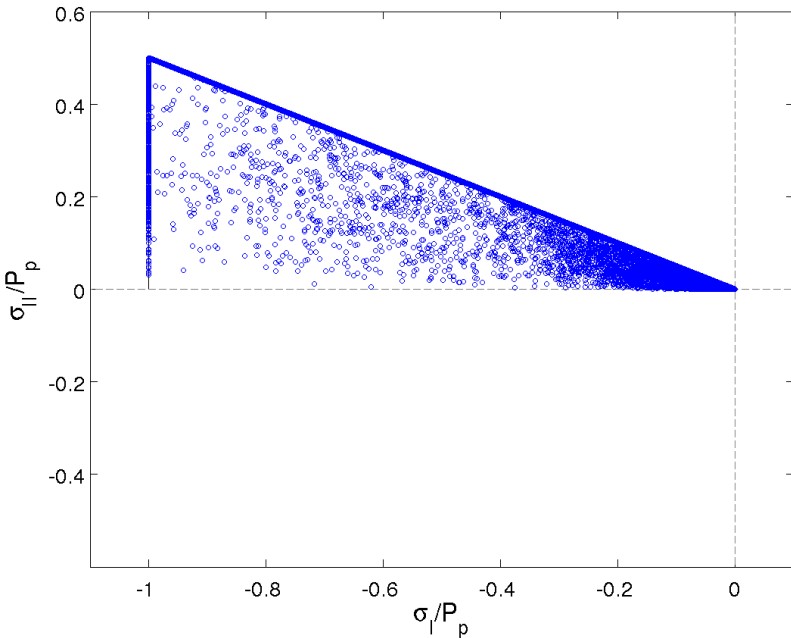

**Figure 4.** Stress invariants after two nonlinear iterations calculated only from $\mathbf{u}^k$ and normalized by the ice strength $P_p$. The solution appears to be numerically converged because the $\sigma_{ij}$ are only a function of $\mathbf{u}^k$. The yield curve (solid black line) is based on a Mohr-Coulomb failure criterion with compressive capping.

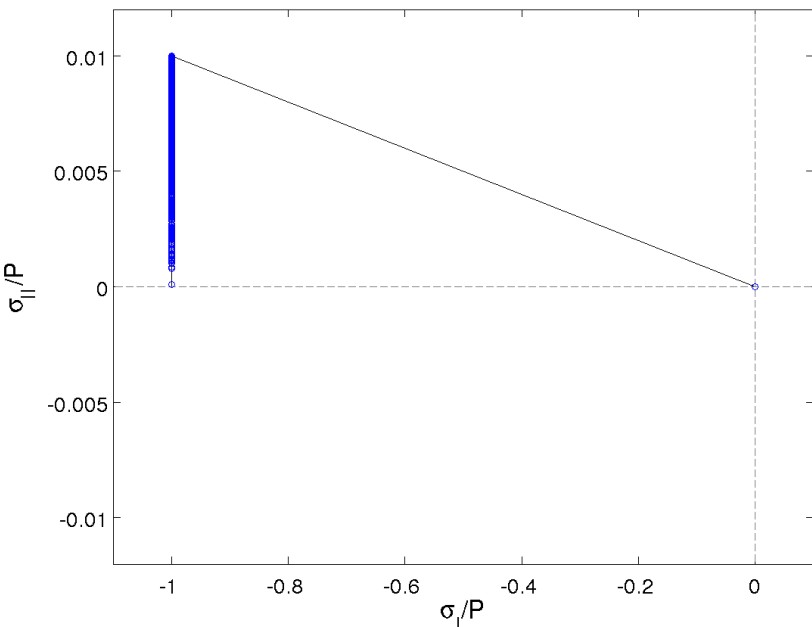

**Figure 5.** Fully converged stress invariants normalized by the replacement pressure $P$. The yield curve (solid black line) is based on a Mohr-Coulomb failure criterion with compressive capping.