# Peer review of "On the calculation of normalized viscous-plastic sea ice stresses"

_Geoscientific Model Development, 2019_

## Referee Comment (RC1) · Anonymous Referee #1 · 9 Dec 2019

The note "On the calculation of normalized viscous-plastic sea ice stresses" by Lemieux and Dupont describes how to compute normalised viscous-plastic sea ice stress properly. They also describe two common traps one can fall into when computing this quantity. This is a valuable (small) contribution that would have saved me from trying to figure out things myself (and wasting a lot of time on that). The text is clearly written, there are a few small comments to consider, see below. The representation is convincing and the explanation of the procedure and the common errors are clear.

I have one small ussue. I would like the authors to revisit the derivation of their equation (6). First, one needs eqs(1,3,4,5) (and not just 1 and 5) and Delta to arrive at an expression like this; second, it only works if P_p in eq(1) is replaced by the replacement pressure P (that's not immediately clear from the text). If one does not want to use

the replacement pressure P (and there are reasons to do so), the derivation ends up with with P_p instead of P on the rhs, because in eq(1) P_p is on the rhs. This is important because eq(10) with then have a "1" instead of P/P_p and in eq(16) it would be P_p/P instead of "1". This has implications for the interpretation (but not for the general conclusions, as far as I can see). Adding a treatment of the no-replacement pressure case would be very helpful for the generality of the paper, so I recommend that the paper be published only after addressing this issue.

Minor comments and suggestions:

page 1 l21 large spatial

l24 Unfortunately, . . . I would add how that leads to misunderstandings in order to formulate a "problem statement". If we all assume we know what we are doing then there's no problem. E.g., Subtle mistakes in calculating stresses can lead to a complete misinterpretation of the state of convergence. Or similar . . .

page 2 l40: I prefer to write Delta als sqrt( (e11+e22)ˆ2 + eˆ{-2}((e11-e22)ˆ2+4e12ˆ2) ), because it is also more straightforward to implement . . .

page 3 l63: such as a Picard solver . . . ow with a Newton solver

l65: Kimmritz et al 2015 use the terminology of "modified" EVP. "revised" EVP was used by Bouillon et al 2013.

l76 a Picard solver

page 5 l117: remove: that could be done by modelers

l122: truely?

page 6 l140: remove "that could be made by modelers"

l145 rephrase sentence: This is the equation of an ellipse we obtain if the principal stresses are normalized by the replacement pressure.

l149, but why only for the elliptical yield curve and not for th Coulombic and Diamond yield curves?

page 7 l166: gives

page 10 Figure 2: I think the caption is misleading. It should start with the statement that sigma is computed based on uˆk only.

---

## Referee Comment (RC2) · Anonymous Referee #2 · 19 Dec 2019

Topic of the manuscript is the correct evaluation of the normalized viscous-plastic sea ice stresses. The model under consideration is the viscous-plastic sea ice model which was formulated by Hibler in 1979 with a replacement pressure introduced by Kreyscher et al. in 2000. The manuscript focuses on the evaluation of the normal stress that are archived with a Picard solver. Two error sources that may occur using the diagnostic are described.

Main issues:

I miss a more detailed discussion of the term numerical convergence of the VP solution and a more careful use of the term numerical convergence. Sometimes you describe by numerical convergence that all stress states are on/in the ellipse ( physical consistency) sometimes you use the term the numerical convergence for the convergence of the sea

ice velocity. Please distinguish better between this two cases. Applications might think that the diagnostic implies numerical convergence of the solution VP solution (sea ice velocity). Explanation/motivation why plotting the normalized stresses is a suitable diagnostic to evaluate numerical convergence of the VP solution in a first step. Clarify that being physical consistent does not imply that one has a convergent approximation of the sea ice velocity. Maybe add a paragraph to the introduction how this diagnostic needs to be used.

2) Can you please explain how the diagnostic should be evaluated for Newton-like solvers? I don't think that it is straight forward. Using your 1D example a fully implicit discretized rheology reads as sigma=Pp/(2|epsilon_{k}| epsilon_k) -Pp/2= -Pp. Does this mean that the diagnostic is unnecessary? I do not think so as Newton-type methods also introduce some from of linearization...

3) Please provide the explicit formulation of the yield curve that you use to plot the figures.

4) Is the diagnostic effected if other limitations are used in (4)? How to deal with different linearization

I recommend that the paper be published only after addressing this issues.

Minor issues:

L. 5 -8 The first example is true for approximations calculated with Picard solver. What about Newton and EVP? The 2 sentences can be misleading

L106 Here numerical solution describes the numerical convergence of v. In line 90 the term numerical convergence is used to describe that the stress states are in/on the ellipse (which is the physical consistency). Be more specific when using the term numerical convergence.

L106 The residual of the momentum equation? Which residual ?

L107-110 I think this point must be emphasized and moved to the introduction ( see main issue 1))

L 121 The solution of the momentum equation ?

L154 Please be more specific how numerical convergence can be assessed

---

## Author Comment (AC1) · 28 Jan 2020

**Response to reviewer 1**

We would like to thank reviewer 1 for his/her very helpful comments. Based on the reviewers' comments we realized that some clarification was needed. We have therefore added two new sections ("The normalized yield curve" and "Broader considerations") and we have also written explicitly the steps that should be followed for calculating and plotting the normalized VP stresses. We have adressed these comments with the goal of keeping the manuscript relatively short. Indeed, we want this manuscript to be a "quick" guide for sea ice modelers. Below, the comments from the reviewers (1) are in normal character. Our responses (2) are in bold

while changes to the manuscript (3) mentioned here are also in bold and in quotes. Note that modifications in the revised manuscript are shown in magenta.

REVIEWER 1

(1) The note "On the calculation of normalized viscous-plastic sea ice stresses" by Lemieux and Dupont describes how to compute normalised viscous-plastic sea ice stress properly. They also describe two common traps one can fall into when computing this quantity. This is a valuable (small) contribution that would have saved me from trying to figure out things myself (and wasting a lot of time on that). The text is clearly written, there are a few small comments to consider, see below. The representation is convincing and the explanation of the procedure and the common errors are clear.

I have one small issue. I would like the authors to revisit the derivation of their equation (6). First, one needs eqs(1,3,4,5) (and not just 1 and 5) and Delta to arrive at an expression like this; second, it only works if $P_p$ in eq(1) is replaced by the replacement pressure $P$ (that's not immediately clear from the text). If one does not want to use the replacement pressure $P$ (and there are reasons to do so), the derivation ends up with $P_p$ instead of $P$ on the rhs, because in eq(1) $P_p$ is on the rhs. This is important because eq(10) with then have a "1" instead of $P/P_p$ and in eq(16) it would be $P_p/P$ instead of "1". This has implications for the interpretation (but not for the general conclusions, as far as I can see). Adding a treatment of the no-replacement pressure case would be very helpful for the generality of the paper, so I recommend that the paper be published only after addressing this issue.

**(2) Ok it is now mentioned in the revised manuscript that equations (1,3,4,5) are needed.**

**Setting $P = P_p$ leads to the following equation**

$$\left(\frac{\Delta}{\Delta^*}\right)^2 = \left[\sigma_{p1}^n + \sigma_{p2}^n + \frac{P_p}{P_p}\right]^2 + e^2 \left[\sigma_{p1}^n - \sigma_{p2}^n\right]^2. \tag{1}$$

**This slightly different equation (compared to equation(10) in the manuscript) does not affect the recommendations given here. The ratio $\frac{P_p}{P_p}$ or $\frac{P}{P_p}$ (equation(10)) defines the center of the ellipse. As mentionned by Geiger et al. 1998, normalized states of stress in the viscous regime are positionned on concentric ellipses all centered at $(-0.5, 0.5)$ when $P = P_p$ while the center of an ellipse when using the replacement pressure depends on the ratio $\frac{P}{P_p}$. The following sentence has been added to the revised manuscript (in the new section called "Broader considerations").**

**(3) "If one does not use a replacement pressure, the stresses in step 2 should be calculated the same way with $P = P_p$. Instead of lying on ellipses defined by equation (10), the normalized viscous states of stress would lie on concentric ellipses centered at $\sigma_{p1}^n = \sigma_{p2}^n = -0.5$ (Geiger et al. 1998)."**

Minor comments:

(1) page 1 l21 large spatial

**(2) Done.**

(1) l24 Unfortunately,...I would add how that leads to misunderstandings in order to formulate a "problem statement". If we all assume we know what we are doing then there's no problem. E.g., Subtle mistakes in calculating stresses can lead to a complete misinterpretation of the state of convergence. Or similar...

**(2) We have added the following sentence:**

**(3)"As demonstrated here, care must be taken when calculating the normalized stresses as two potential mistakes can lead to a misinterpretation of modeling results."**

(1) page 2 l40: I prefer to write $\Delta$ as $((e_{11} + e_{22})^2 + e^{-2}((e_{11} - e_{22})^2 + 4e_{12}^2))^{\frac{1}{2}}$, because it is also more straightforward to implement

**(2) We agree. It has been changed.**

(1) page 3 l63: such as a Picard solver...or with a Newton solver

**(2) Done.**

(1) l65: Kimmritz et al 2015 use the terminology of "modified" EVP. "revised" EVP was used by Bouillon et al 2013.

**(2) Ok it has been corrected.**

(1) l76 a Picard solver

**(2) We decided to keep the sentence as is because the words solve, solution and solver are already present...**

(1) page 5 l117: remove: that could be done by modelers

**(2) Done.**

(1) l122: truely?

**(2) We don't think we need to add 'truely'.**

(1) page 6 l140: remove "that could be made by modelers"

**(2) Done.**

(1) l145 rephrase sentence: This is the equation of an ellipse we obtain if the principal stresses are normalized by the replacement pressure.

**(2) Ok it has been rephrased.**

(1) l149, but why only for the elliptical yield curve and not for the Coulombic and Diamond yield curves?

**(2) This is a good question...We argue that the recommendations in our manuscript apply to all the yield curves. We do not have the diamond nor the modified coulombic yield curve implemented in the McGill model but we have recently coded a standard Mohr-Coulomb yield curve with compressive capping (i.e., a triangle). This other constitutive formulation is obtained by writting $\zeta$, $\eta$ and $P$ as**

$$\zeta = \frac{P_p}{2|\dot\epsilon_I^*|}, \tag{2}$$

$$P = 2\zeta|\dot\epsilon_I|, \tag{3}$$

$$\eta = \frac{\left(\frac{P}{2} - \zeta\dot\epsilon_I\right)\sin\phi}{2\dot\epsilon_s^*} \tag{4}$$

where $\dot\epsilon_I$ is the divergence, $|\dot\epsilon_I^*| = \max(|\dot\epsilon_I|, d_{min})$ with $d_{min}$ a small number similar to $\Delta_{min}$, $\phi$ is the angle of friction, $\dot\epsilon_s^* = \max(\dot\epsilon_s, s_{min})$ with $\dot\epsilon_s$ the maximum shear strain rate and $s_{min}$ another small number, set equal to $d_{min}$.

Note that this formulation of Mohr-Coulomb assumes a pure shear flow rule. Divergence (larger than $d_{min}$) can only occur at the tip of the triangle and convergence when the sea ice pressure is equal to $P_p$.

It is observed that with this new rheology, the Picard solver really struggles to obtain a numerically converged solution. With $P* = 27.5 \times 10^3$ **Nm**$^{-2}$ , $d_{min} = 2 \times 10^{-9}$ **s**$^{-1}$ and $sin\phi = 0.5$ (i.e., $\phi = 30°$), the solver does not converge. When calculating the normalized stresses the proper way, there are states of stress outside the yield curve. Similar to the results obtained with the elliptical yield curve, the stresses (shown in Fig.1 in this document) appear to have converged if only $\mathrm{u}^k$ is used to calculate the normalized stresses.

To obtain a fully converged solution, some rheology parameters were modified from the values given above; $P* = 5 \times 10^2$ **Nm**$^{-2}$, $sin\phi = 0.01$ and $d_{min} = s_{min} = 2 \times 10^{-8}$

s$^{-1}$. **Consistent with the results obtained with the ellipse, the fully converged stresses normalized by $P_p$ are either on or inside the yield curve (shown in Fig.2).**

**Fig.3 shows the fully converged stress invariants when normalized by the replacement pressure $P$. As for the ellipse, the solution is not realistic as there are no stresses in the viscous regime. Strangely, there are no states of stress on the long side of the triangle. This can understood when considering the normalized first stress invariant ($\sigma_I = (\sigma_{11} + \sigma_{22})/2$). It is easy to show that**

$$\sigma_I = \zeta \left( \dot{\epsilon}_I - |\dot{\epsilon}_I| \right). \tag{5}$$

**Normalizing by $P = 2\zeta|\dot{\epsilon}_I|$, the normalized $\sigma_I$ (i.e. $\sigma_I^n$) is**

$$\sigma_I^n = \frac{1}{2} \left( \frac{\dot{\epsilon}_I}{|\dot{\epsilon}_I|} - 1 \right). \tag{6}$$

**Consistent with what is observed in Fig.3, $\sigma_I^n$ can take only two possible values: $\sigma_I^n = 0$ if $\dot{\epsilon}_I > 0$ (divergence) or $\sigma_I^n = -1$ if $\dot{\epsilon}_I < 0$ (convergence).**

**To further support our conclusions, we have added some of this material in the revised manuscript (in the new section "Broader considerations".**

**Going back to the results of Wang and Wang 2009, it seems that the diamond yield curve does not use a replacement pressure (see their p.3). We don't know, however, why some states of stress are inside the yield curve for the modified Coulomb. Is it possible it was correctly normalized by the ice strength? This is not discussed in the manuscript but we speculate that Wang and Wang also made the other mistake: they calculated the normalized stresses only using the latest iterate. This problably explains**

why the solution seems to have converged. Consistent with our new results in our revised manuscriopt when using a Mohr-Coulomb, Ringeisen et al. 2019 were not able to get numerical convergence with the modified Coulomb (see their Fig. 12). Anyway, we think that mentioning this figure from Wang and Wang only adds more confusion as it is not clear what they did exactly. We have removed the reference to it in the revised manuscript.

(1) page 7 l166: gives

**(2) Done.**

(1) page 10 Figure 2: I think the caption is misleading. It should start with the statement that sigma is computed based on $u^k$ only.

**(2) Done.**

**Jean-François Lemieux**

**REFERENCES**

**Geiger,C.A., W.D. Hibler and S.F. Ackley, "Large-scale sea ice drift and deformation' Comparison between models and observations in the western Weddell Sea during 1992", J. Geophys. Res, 103, 21893-21913, 1998.**

**Ringeisen, D., M. Losch, B. Tremblay and N. Hutter, "Simulating intersection angles between conjugate faults in sea ice with different viscous-plastic rheologies", The Cryosphere, 13, 1167-1186, 2019.**

[Figure]

norm_stress_inv_MC_73.png

norm_stress_inv_MC_71.png

`norm_stress_inv_MC_72.png`

---

## Author Comment (AC2) · 28 Jan 2020

**Response to reviewer 2**

We would like to thank reviewer 2 for his/her very helpful comments. Based on the reviewers' comments we realized that some clarification was needed. We have therefore added two new sections ("The normalized yield curve" and "Broader considerations") and we have also written explicitly the steps that should be followed for calculating and plotting the normalized VP stresses. We have adressed these comments with the goal of keeping the manuscript relatively short. Indeed, we want this manuscript to be a "quick" guide for sea ice modelers. Below, the comments from the reviewers (1) are in normal character. Our responses (2) are in bold

while changes to the manuscript (3) mentioned here are also in bold and in quotes. Note that modifications in the revised manuscript are shown in magenta.

REVIEWER 2

(1) Topic of the manuscript is the correct evaluation of the normalized viscous-plastic sea ice stresses. The model under consideration is the viscous-plastic sea ice model which was formulated by Hibler in 1979 with a replacement pressure introduced by Kreyscher et al. in 2000. The manuscript focuses on the evaluation of the normal stress that are archived with a Picard solver. Two error sources that may occur using the diagnostic are described.

Main issues:

(1) I miss a more detailed discussion of the term numerical convergence of the VP solution and a more careful use of the term numerical convergence. Sometimes you describe by numerical convergence that all stress states are on/in the ellipse ( physical consistency) sometimes you use the term the numerical convergence for the convergence of the sea ice velocity. Please distinguish better between this two cases. Applications might think that the diagnostic implies numerical convergence of the solution VP solution (sea ice velocity). Explanation/motivation why plotting the normalized stresses is a suitable diagnostic to evaluate numerical convergence of the VP solution in a first step. Clarify that being physical consistent does not imply that one has a convergent approximation of the sea ice velocity. Maybe add a paragraph to the introduction how this diagnostic needs to be used.

**(2) The stresses are function of the velocity vector. A numerically converged velocity therefore leads to numerically converged stresses (consistent with the physics of the problem). We have improved the text and introduce the nonlinear residual vector to**

better explain what we mean by numerical convergence. We have also added in section 4 the sentence:

(3)"A 'fully' converged solution for $u$ is characterized by a small residual. As the stresses are function of $u$, a fully converged velocity vector leads to states of stress that are either on (plastic) or inside (viscous) the yield curve."

(2) We also better explain in the introduction why the normalized stresses are useful to assess convergence and physical consistency. The last paragraph of the introduction now starts with the following sentences:

(3) "Calculating and plotting the normalized states of stress with respect to the yield curve is a useful diagnostic for assessing the physical consistency and numerical convergence of a VP solution. Indeed, this method can confirm whether a sea ice rheology is properly implemented in a model. The method is also helpful for evaluating numerical convergence. This is especially true for the explicit elastic-VP (EVP) solver (e.g., Hunke, 2001) which does not include a measure of convergence such as a residual."

(1) Can you please explain how the diagnostic should be evaluated for Newton-like solvers? I don't think that it is straight forward. Using your 1D example a fully implicit discretized rheology reads as $\sigma = Pp/(2|\epsilon_k|\epsilon_k) - Pp/2 = -Pp$. Does this mean that the diagnostic is unnecessary? I do not think so as Newton-type methods also introduce some form of linearization...

(2) The same method should be followed for Newton solvers. Indeed, they are also based on a linearization with $u^{k-1}$ for the calculation of the Jacobian matrix and the residual vector.

(1) Please provide the explicit formulation of the yield curve that you use to plot the figures.

**(2) the formulation has been written in the captions of Fig.1-3. The normalized yield curve is also better explained in section 3 of the revised manuscript. The following text has been added after equation (10):**

**(3) "...which describes a family of ellipses that depend on the ratio $\Delta/\Delta^*$ for their size and on the ratio $P/P_p$ for their center. Equation (10) with $\Delta/\Delta^* = P/P_p = 1$ defines what we refer to as the normalized yield curve in principal stress space. Hence, according to our rheology, normalized plastic stresses should fall on the normalized yield curve while normalized viscous stresses should lie on smaller ellipses inside the normalized yield curve (Geiger et al. 1998)."**

(1) Is the diagnostic effected if other limitations are used in (4)? How to deal with different linearization?

**(2) Good points. We have added in the revised manuscript a section called "Broader considerations" where we argue that our conclusions remain the same for other approaches for the limitation (e.g. Kreyscher et al. 2000, Lemieux and Tremblay 2009):**

**(3) "The recommendations given above remain the same if another approach is used for limiting the viscous coefficients (see equation 4). Numerical experiments with the approach of Kreyscher et al. 2000 or with the hyperbolic tangent of Lemieux and Tremblay 2009 allow one to draw the same conclusions (not shown)."**

**(2)** We have also added a sentence to explain how the normalized stresses should be calculated when using another kind of linearization (e.g. Lemieux and Tremblay 2009):

**(3)** "While it is not recommended to linearize the rheology term with the previous two iterates (as done by Lemieux and Tremblay 2009) the stresses in step 2 (see beginning of section 5) should in this case be obtained from $\sigma_{ij} = 2\eta(\mathbf{u}_l)\dot{\epsilon}_{ij}(\mathbf{u}^k) + [\zeta(\mathbf{u}_l) - \eta(\mathbf{u}_l)]\dot{\epsilon}_{kk}(\mathbf{u}^k)\delta_{ij} - P(\mathbf{u}_l)\delta_{ij}/2$ with $\mathbf{u}_l = (\mathbf{u}^{k-1} + \mathbf{u}^{k-2})/2.$"**

(1) I recommend that the paper be published only after addressing these issues.

Minor issues:

(1) L. 5 -8 The first example is true for approximations calculated with Picard solver. What about Newton and EVP? The 2 sentences can be misleading.

**(2) Thanks for pointing this out. The same conclusions apply for both Picard and Newton solvers (see comment above). We have, however, clarified the text in the abstract and in the conclusion to clearly state that the first "mistake" can only be made for implicit solvers while the second one applies to implicit and explicit solvers.**

(1) L106 Here numerical solution describes the numerical convergence of v. In line 90 the term numerical convergence is used to describe that the stress states are in/on the ellipse (which is the physical consistency). Be more specific when using the term numerical convergence.

**(2) See our response to your first main comment above.**

(1) L106 The residual of the momentum equation? Which residual?

**(2) The residual $F = Au - b$ and the criterion for the nonlinear convergence have been introduced in the revised manuscript. This sentence has been modified.**

(1) L107-110 I think this point must be emphasized and moved to the introduction ( see main issue 1))

**(2) This is a conclusion of Lemieux and Tremblay 2009. We prefer not to repeat it elsewhere in the manuscript. However, the sentence was not well formulated and we have therefore decided to rephrase it:**

**(3) "Note that, in general, the fact that states of stress are on or inside the yield curve does not imply full convergence; the final positions (on and inside the yield curve) are obtained once $u^k$ is the fully converged solution (Lemieux and Tremblay 2009)."**

(1) L 121 The solution of the momentum equation?

**(2) See our response to your first main comment above.**

(1) L154 Please be more specific how numerical convergence can be assessed.

**(2) See our other responses above.**

**Jean-François Lemieux**

**REFERENCES**

Geiger,C.A., W.D. Hibler and S.F. Ackley, "Large-scale sea ice drift and deformation'
Comparison between models and observations in the western Weddell Sea during 1992",
J. Geophys. Res, 103, 21893-21913, 1998.

Interactive
comment